# Incorporating Social and Policy Drivers into Land-Use and Land-Cover Projection

**Behnoosh Abbasnezhad** [1] , **Jesse B. Abrams** [1,2,*] and Jeffrey Hepinstall-Cymerman [1]

1   Warnell School of Forestry and Natural Resources, University of Georgia, Athens, GA 30602, USA; behnoosh@uga.edu (B.A.); jhepinstall@warnell.uga.edu (J.H.-C.)
2   Savannah River Ecology Laboratory, University of Georgia, Athens, GA 30602, USA
*   Correspondence: jesse.abrams@uga.edu; Tel.: +1-404-981-1707

**Abstract:** Forestlands in the southeastern U.S. generate a great variety of ecosystem services that contribute to the well-being of humans and nonhumans alike. Despite their importance, forests continue to be lost to other land uses such as agricultural production and urban development. Advancements in remote sensing and machine learning techniques have facilitated land use/land cover (LULC) change projections, but many prior efforts have neglected to account for social and policy dimensions. We incorporated key socio-economic factors, conservation policies, societal preferences, and landscape biophysical features into LULC projection techniques under four different development scenarios. We applied this approach in the Upper Flint watershed, which flows south from the Atlanta, Georgia metropolitan area and is characterized by extensive urbanization and associated deforestation. Our results suggest that incorporating social and policy drivers in future LULC projection approaches leads to more realistic results with higher accuracy levels, offering decision-makers, development planners, and policymakers better opportunities to forecast the effects of anticipated changes on the availability of ESs in the future. Conservation organizations and public agencies can benefit from such analysis to identify regions requiring conservation interventions for prioritizing their conservation efforts. We used publicly available data for the conterminous U.S., hence our approach can be replicable in other study regions within the nation.

**Keywords:** LULC projection; urbanization; deforestation

## 1. Introduction

Anthropogenic activities are the principal drivers of change in terrestrial ecosystems [1]. Land cover and land management changes affect exchanged energy, water, aerosol, and greenhouse gas (GHG) fluxes between the land and atmosphere [2]. Alteration of land use and land cover (LULC) patterns can significantly affect natural ecosystem functioning and the ecosystem services (ESs) they provide to human societies, not only at a local scale, but also through GHG fluxes and changes in radiative transfer contributing to changes in atmospheric chemistry and thermal and moisture balance at global scales [2]. Growing concern over the impacts of LULC changes on ESs has prompted scientists, decision-makers, policymakers, and other stakeholders to use LULC projections in forecasting and exploring the effects of anticipated changes on the availability of ESs [3]. Simulating ESs can be accomplished via LULC projections that model current and alternative trajectories and help to identify the most vulnerable ecosystems for sustainable land management practices or conservation purposes. These projections can also serve to evaluate the costs and benefits of varied land management decisions [4].

Advancements in remote sensing, GIS capabilities, and machine learning techniques [5] have allowed for the development of varied projection models that are diverse in their conceptual foundations, thematic foci, spatial characteristics, and modeling methodologies. Cellular automation (CA) in conjunction with Markov chain (MC) analysis has historically



been one of the most popular approaches [6] and has been used in diverse regions to analyze past, present, and future land-use changes [7–10]. This approach incorporates both spatial and temporal components of land-cover dynamics in detecting past and simulating future land-use change patterns [11]. Moreover, projections from such models have been applied to model changes in various ecosystem services [3,12–16]. However, general models, including many that employ CA-MC processes, fail to explain transitions as a function of human decision-making, despite the fact that LULCs are continually affected by a variety of human-driven forces [10,17].

As it is desirable to use available models with maximum generality, realism, and precision of understanding and predicting systems [18], in this study we aim to combine a CA-MC model with other relevant variables to calculate land-use transition possibilities for the Upper Flint River Watershed near Atlanta, Georgia. We use publicly available and spatially explicit data for the conterminous United States to run our projection models, once with key socio-political variables and once without them, to determine whether incorporating these factors drives meaningful change in projected future land uses. Based on this analysis, we draw conclusions regarding the value of incorporating social, political, and economic variables in LULC projection models.

## 2. Background

### 2.1. Drivers of Land-Use Change in U.S. Urban Areas

The anthropogenic driving forces of LULC change include demographic factors, affluence, political structure, economic factors, land tenure patterns, and landowners' attitudes and values [17]. These driving forces can change over time for many reasons, including in response to economic cycles, infrastructure development, policy change, and changes in social attitudes [19,20]. For example, since the mid-20th century, many American metropolitan areas have experienced remarkable growth in population and associated development. In many cases this has been characterized by expanding urbanization, spreading outward from an urban core towards the suburbs and exurbs such that, in many regions, urban expansion rates exceed net population growth rates [21]. The conversion of undeveloped land to residential, industrial, and associated uses not only alters landscapes but can result in the loss of important ESs as forests and other natural or semi-natural land covers are replaced by the built environment [22].

Forestlands generate a wide range of ESs that contribute to human well-being, including carbon sequestration, nutrient retention, water purification, drought and flood mitigation, habitat provisioning for a variety of species, and timber production [23–26]. Forest conversion to other land uses such as agricultural production and urban development alters the provisioning of these ESs [27]. Conversion to urban development has been reported as an especially acute threat to the availability of provisioning ESs [28,29]. These ES losses occur as the result of soil erosion, lower water and food supplies, higher urban storm water runoff, reduced carbon stocks, fragmented landscapes, and diminished biodiversity [24,30,31].

While some drivers of LULC change operate on decadal time scales, others develop much more rapidly [32]. For example, in the US the coronavirus (COVID-19) pandemic that began in 2020 not only affected public health, but also prompted many residents to seek a new lifestyle characterized by seclusion from crowded public places and the use of personal in place of public transportation. According to the US Census Bureau, e-commerce sales increased by 43% in 2020. The demands associated with the pandemic accelerated a trend toward remote job opportunities and remote or hybrid options for university and graduate coursework, all of which affected residential patterns and associated markets. The pandemic also triggered substantial declines in income, increases in unemployment, and a long-lasting impact globally in many sectors including energy, transportation and logistics, and manufacturing industries [33,34]. At the same time, it accelerated preexisting trends favoring suburban and exurban development in areas of high domestic in-migration [19].

## 2.2. Public Policy Drivers and Constraints on LULC

The public policy system for natural resources in the U.S. includes multiple, often overlapping, policy tools at federal, state, county, and municipal levels that range from mandatory to voluntary in their application. Examples of mandatory policies include the Endangered Species Act and the Clean Water Act. The Endangered Species Act prohibits any activity that could directly or indirectly harm any species listed as "endangered" [35] and restricts many activities that could harm species listed as "threatened". The Clean Water Act relies upon states, federally recognized tribes, and territories to manage non-point source pollution (pollution that results from activities across the landscape such as farming, forestry, construction, and associated runoff). Non-point source management often occurs through voluntary programs, but enforcement mechanisms exist to ensure that states are not overly permissive in their water quality regulation.

In addition to mandatory policies, there is a growing trend toward voluntary and market-based policy mechanisms that use economic incentives in place of threats of punishment. Examples include preferential taxation of undeveloped forestland, the use of conservation easements (sales of development rights that leave the other core ownership rights intact), state and federal cost-share programs, economic support for wood-based bioenergy, and conservation land purchases from willing sellers to incentivize forest conservation efforts [36]. The majority of forestland in the U.S. is owned and managed by private owners including individuals, families, trusts, estates, and corporations; family forest owners (formerly termed "non-industrial forest owners") own more acreage in aggregate than do corporate owners. Research shows that ownership motivations for family forest owners go well beyond maximizing timber production to include factors such as "beauty or scenery", "protection or improvement of wildlife habitat", and "recreation" [37]. Considering such diverse motivations, incentive-based policies can encourage private forest owners to maintain their forestlands and receive technical assistance, annual payments, or reductions in tax obligations in return [38,39].

## 2.3. Integrating Social, Economic, and Policy Factors into LULC Modeling

LULC change modeling based on CA-MC uses historical land-use changes as a guide to future scenario development [40]. Under this approach, at least two LULC maps with a time interval are used to assess LULC transition patterns. Transitions in land cover in each pixel in future time steps are then calculated based on past land-use trends [41], the spatial neighborhood of each pixel, or a desired or predicted land-cover abundance at a broader scale [3]. Although these models can be spatially detailed and ecologically meaningful in predicting future LULCs and associated ES effects [40], there are numerous anthropogenic forces that should be incorporated into these models to help land managers anticipate potential effects of LULC changes on a variety of ecological and social processes [27]. Future LULC projection models that lack landowner decision-making processes and policy incentives and disincentives cannot be expected to simulate future LULCs realistically [42]. Hence, some scientists have worked to develop more reliable models to project future LULC patterns by including econometric factors (e.g., Gross Domestic Production, income, poverty rate) [16,43–45] or by combining biophysical features and voluntary and regulatory policies [3,10,27]. The current study builds upon previous attempts to develop a realistic model to project future LULCs by incorporating prominent socio-economic factors, mandatory and voluntary policies, societal preferences, and landscape biophysical features within a CA-MC approach. For this study we used publicly available data for the entire conterminous U.S. so that our approach can be replicable in other study regions within the nation.

## 3. Study Area

Georgia, located in the southeastern U.S., is one of the nation's fastest-growing states [46]. The Atlanta Metropolitan Area (AMA) is experiencing one of the highest population growth rates in the country due to a combination of natural increase and both domestic and international in-migration [47]. The AMA had a population of approximately

4.95 million people in 2020 and the population is projected to exceed 6 million by 2040 [48]. The Upper Flint watershed (UFW), located in the southern part of the AMA, is an important watershed in terms of providing habitat for many endangered and threatened species, water resources for agriculture and personal consumption, and recreational opportunities. The Flint is Georgia's second longest river and is one of the rivers included in the tri-state water dispute among the states of Georgia, Alabama, and Florida regarding flows in the Apalachicola-Chattahoochee-Flint (ACF) River Basin [49]. With an area of 682,188 hectares, the UFW covers portions of 19 counties. Larger cities located in this region include College Park, Fairburn, Fayetteville, Newnan, Peachtree City, Riverdale, Tyrone, Union City, Manchester, and Woodberry [49,50]. The northern extent of the UFW is traversed by over 46 miles of major transportation corridors and is the site of Hartsfield-Jackson Atlanta International Airport and its supporting businesses. The Flint River, Camp Creek, and Morning Creek watersheds in southeastern Fulton County and northwestern Clayton County also include a relatively large amount, 15 percent, of high- or medium-density residential lands [49]. The Flint River and its tributaries provide numerous provisioning ESs such as water for agricultural, industrial, and municipal uses along with varied regulating, supporting, and cultural ESs; an example of the latter is recreational opportunities for local people. The watershed is home to a diverse population of flora and fauna including several species that are listed as endangered or threatened [51].

Extensive urbanization, inadequate stormwater controls, and deforestation have limited provisioning of some important ESs within the watershed [49,50]. Recent local reports note an alarming decline in groundwater levels and streamflow coupled with increasing water demands, leading to reduced water availability and increased pressure on water resources [51,52]. Additionally, the region has been experiencing increases in average annual temperatures and more frequent and intense precipitation events, which may have implications for water quality and ecosystem health [51].

The elevation of the watershed ranges from 82 m to 410 m with an average of 212 m above sea level [53]. Average annual precipitation from 2001 to 2021 is 1337 mm and the mean annual temperature is 17.3 °C [54]. More than half of the UFW is covered by forestlands; 25% of the watershed is categorized as evergreen forest, 20.3% as deciduous forest, and 6.3% as mixed forest [55] (Figure 1).

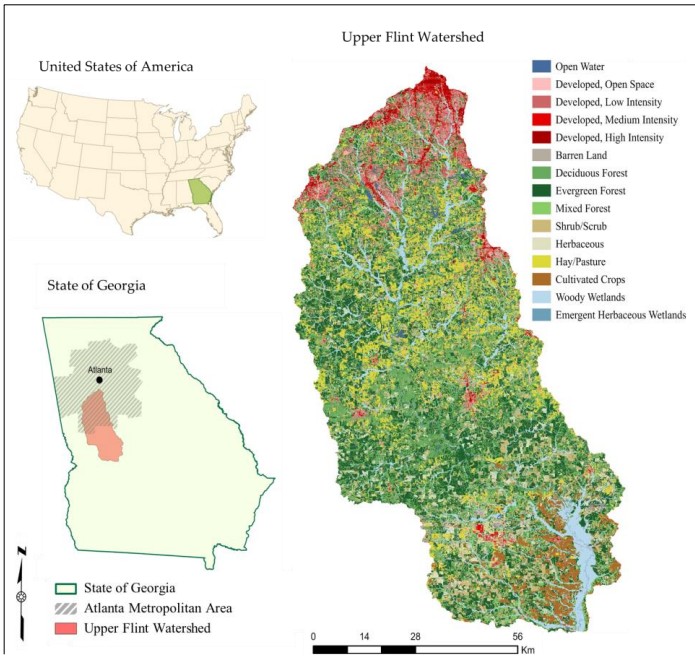

**Figure 1.** The Land-use and Land-cover map of the Upper Flint Watershed in 2019 based on National Land Cover Database (NLCD) produced by the U.S. Geological Survey.

## 4. Methods

This section is divided to four main parts. The first part describes our approach to analyzing patterns of land use transition within the UFW between 2011 and 2019 as a means of projecting changes through 2040. The second part explains how conservation policies and socio-economic variables were incorporated into our LULC projection approach. The third part defines potential assumptions for future developments and patterns of LULC change in the UFW under four various development scenarios, and the final part is dedicated to the model validation process.

### 4.1. Land-Use and Land-Cover Projection

The module Land Change Modeler (LCM), developed as an empirically parameterized land change projection tool embedded in the software TerrSet 19.0.7, was utilized to project a LULC map for the UFW for the year 2040. The application of LCM across many disciplines in varied geographic areas since its introduction in 2006 is well documented [10,27,56]. LCM uses two LULC maps from distinct periods as input data to analyze transitions among land-cover classes. We used National Land Cover Database (NLCD) maps with 30m spatial resolution (released every 5 years for the conterminous United States) as input data [55]. We used the 2001, 2006, 2011, 2016, and 2019 maps to generate the baseline for projecting the 2040 LULC map for the UFW. Georgia Regional Commissions have prepared their future land management and development plans; hence we utilized the same time frame to incorporate local governments' strategic planning timeline into our model.

To reduce the potential combination of transition sub-models and improve accuracy, we modified the number of NLCD land-use classes from 15 to 9 (for more information see the Supplementary, Table S1). We first used the 2011 and 2016 NLCD maps to project LULC in 2019 using the LCM module and compared our projected map with the NLCD 2019. Further details of this step are provided below in the Model Validation section. As the initial simulation showed over 90 percent accuracy level, we ran the projection for 2040 using the same process.

The initial historical land-use change assessment results represented 67 transition categories among our 9 LULC classes, ranging from 31,000 ha to less than one hectare in total area. To eliminate unnecessary details and erroneous changes derived from classification errors and to maintain focus on the core objective of this study (projecting the conversion of forestlands to urban areas), we narrowed down transition potentials to 20 categories by including those related to forest or urban areas and larger than 20 ha in size [57]. We then categorized transitions as "Deforestation", "Afforestation", "Pine plantation", and "Urbanization" sub-models. A sub-model is a group of transitions that share similar underlying driving determinants when creating predictions [58].

The next step was to develop a candidate set of predictor variables to calculate the probability of a pixel being converted from one land-cover class to another. These variables can be either static or dynamic components. Static variables are unchanging over time, while dynamic variables are time-dependent and should be recalculated over time during the course of a prediction [57,58]. We set aspect, elevation, slope, and population growth rate layers as our static variables and proximity to urban areas, and proximity to roads, highways and railroads as our dynamic variables. Transition potentials were then created using the Multi-Layer Perceptron (MLP) model, a tool for modeling complex non-linear relationships [57–59].

During the LULC projection process, the model creates a square matrix of land-cover transition probabilities based on an analysis of historical land-cover change; this is known as the MC matrix. The MC model analyzes the probability of LULC changes over time by developing a transition probability matrix. However, Markov analysis does not consider the causes of land-use change, nor is it sensitive to spatial and geographical setting [60,61]. Hence we used the MLP_MC as a combination of the MLP with an MC model for spatiotemporal dynamic modeling [58]. The MLP provides an integration of prominent driving factors of LULC changes under each sub-model while the MC matrix

controls the temporal dynamics of the amount of change in any specific LULC transition [9]. See the Supplementary for examples of MC matrices used in this study.

The LCM generates two projected maps as hard and soft predictions. In hard prediction, a projected map is produced for the prediction year in which each pixel is assigned to a specific land-use class [62]. In soft prediction, the projected map indicates the vulnerability to change, in which each pixel is assigned a value from zero to one. Lower values indicate less vulnerability to change, while higher values indicate high susceptibility to change [62,63].

### 4.2. Constraints and Incentives

Previous studies have suggested that the inclusion of driving forces in the LULC projection process can help to create a more accurate model when compared to LULC projections based solely on MC analysis [10,27,42,64]. TerrSet 19.0.7 software provides the ability to incorporate anthropogenic driving forces into land-use change assessments through the Constraints and Incentives (CI) panel. CI is a raster layer with pixels ranging from 0 to a number slightly above 1. These values reflect land-use change probabilities with regard to socio-economic factors or policies governing land uses. For instance, values of 0 on the map are treated as an absolute development constraint and can represent, for example, strictly protected areas or established conservation easements with no land conversion possibilities, while values above 1 can represent factors promoting conversion of forestlands to other uses. Values less than 1 but above 0 act as disincentives to change while values greater than 1 act as incentives. We considered population growth rate, median household income, regional commission development plans, broadband internet coverage, and land parcelization as development incentives, and the presence of species listed under the Endangered Species Act, wetland and riparian zone protection programs, regional commission conservation plans, established conservation easements, and high conservation value forests (HCVF) as development constraints. See the Supplementary for information about the significance of these CIs and the main sources for acquiring related data.

### 4.3. Future Land Planning Scenarios

The majority of existing LULC projections attempt to determine optimal locations for urban development or intensive agriculture while minimizing negative externalities on other land-use functions [8,16,27,65], to provoke policy discussions based on scientific assessments to help portray the future status of complex ecosystems to identify policy alternatives [10,42,66–68], or to explore potential impacts of LULC changes on ecosystems to inform land-use planning for sustainable economic development and nature conservation [10,66,69]. To be most effective, LULC projections develop multiple alternative scenarios, defined as narratives of potential socioeconomic and environmental trajectories based on starting assumptions [65,69,70]. Scenarios represent "what if" situations, useful in exploring and mapping the consequences of modeled changes in the studied system. Scenarios help to incorporate certain reasonable assumptions such as demographic trends or people's attitudes into LULC projections [67]. In this study, we modeled potential future changes under four main scenarios, each of which used a distinct CI layer (Table 1). While each scenario differs in the effect of constraints and incentives, the population growth rate is held constant across the four scenarios.

### 4.4. Baseline

The baseline or "Business as Usual" scenario assumes that future land-use changes in the UFW follow the same pattern as did transitions between 2011 and 2019. All constraint/incentive factors incorporated into the CI layer are weighted equally and have equal impact on development decision making, except for conservation easements and regional commission conservation plans. The latter were coded as zero in the CI layer (no possibility of conversion), since development in established easements and lands dedicated to regional conservation purposes is assumed to be unlikely.

**Table 1.** The driving forces affecting land-use transitions incorporated into development Constraints and Incentives (CI) layer under four development scenarios. "x2" shows that the contributing factor was given twice the weight in the CI layer calculation; 0 values represent no change possibility.

| | Contributing Factors | Business as Usual | Conservation | Urbanization | Maximum Forest Protection |
|---|---|---|---|---|---|
| | | | Scenarios | | |
| Incentives | Population growth rate | x1 | x1 | x2 | x1 |
| | Median household income | x1 | x1 | x2 | x1 |
| | Regional commission development plan (2040) | x1 | x1 | x1 | x1 |
| | Broadband internet coverage | x1 | x1 | x2 | x1 |
| | Parcelization | x1 | x1 | x2 | x1 |
| Constraints | Presence of species listed under the Endangered Species Act | x1 | x2 | x1 | 0 |
| | Wetland and riparian zone protection | x1 | 0 | x1 | 0 |
| | Protection of HCVFs | x1 | x2 | x1 | 0 |
| | Regional commission conservation plan (2040) | 0 | 0 | x1 | 0 |
| | Established conservation easements | 0 | 0 | 0 | 0 |

*4.5. Urbanization*

This scenario represents a higher urban growth rate than the baseline. Under this scenario, loosening of current conservation policies or greater pressure of market forces for development favor transitions from other land uses to urban areas. Thus, development incentive factors in the CI layer are weighted twice as much as the development constraints. Although some studies suggest testing all pairs of NLCD maps available to identify the time frame with highest urban growth rate to initiate the MC matrix for this scenario [10], we chose to continue using the same MC matrix as for the baseline scenario to avoid biases caused by including large occasional projects such as the Hartsfield-Jackson Atlanta International Airport expansion project and establishment of the McIntosh Dam in the UFW. Unlike in the baseline scenario, areas within regional conservation plans can still be converted to urban development under this scenario. However, established conservation easements remain coded as zero.

*4.6. Conservation*

The conservation scenario places greater emphasis on forest protection and restoration while reflecting a plausible balance between socio-economic and ecological considerations. Under this scenario, the introduction of incentive programs helps owners of forestlands with higher conservation values maintain their properties undeveloped. This scenario also assumes that conservation-based land-use policies focus on riparian zones by increasing the width of buffer areas on main rivers to 100 m and secondary rivers to 30 m and requiring forest protection or reforestation within the buffers. Our reasoning for such an increase in buffer areas is that these areas help protect water bodies from nonpoint pollutants and improve the water quality of streams once restored. In addition, urban development in riparian zones is often costly due to land preparation construction precautions and flooding

risks. As in other scenarios, urban development within buffer areas is prohibited. This scenario used the same MC matrix as the baseline, while in the CI layer, development constraints were weighted twice as much as were development incentives.

### 4.7. Maximum Forest Protection

This scenario portrays the maximum possible forest protection within the UFW, in which, regardless of socio-economic pressures promoting urban sprawl, no existing forest-land would be converted to urban areas. Moreover, forestlands with higher conservation values, and those in riparian zones, would not be converted to any non-forest land uses. While this scenario is not likely from a real-world policy perspective, it does serve as a contrast to the other, more plausible, scenarios.

### 4.8. Model Validation

The validation procedure initially assesses the quality of the predicted LULC map against a reference map to generate Kappa indices that indicate the level of agreement between the two maps [8,9]. Since the use of Kappa indices has become the most prevalent form of accuracy assessment in remote sensing and other fields [71], we first calculated the Kappa indices of agreement $K_{no}$ (overall accuracy of the simulation) and $K_{location}$ (level of agreement of location) [72]. However, considering that there are uncertainties associated with the reliability of Kappa indices, it is suggested that other techniques such as a cross-tabulation matrix with two simpler summary parameters including Quantity Disagreement and Allocation Disagreement be used to summarize and better represent the error matrix [71,73,74]. In this method, the value of a disagreement measure is considered substantial if it exceeds 0.1 [75].

The Validation panel in TerrSet allows the user to determine the quality of the predicted land-cover map in relation to a map of actual land cover by running a 3-way crosstabulation between the later land-cover map, the projected map, and a reference land-cover map [58]. In this study, we ran the LCM module using NLCD 2011 (earlier map) and NLCD 2016 (later map) with and without the baseline CI layer to project a LULC map for 2019. We then assessed the projection accuracy against the NLCD 2019 as the reference map.

## 5. Results

### 5.1. Land-Use Transitions

Comparing the NLCD maps of 2011 and 2019, we see considerable exchanges in Deciduous/Mixed Forest, Evergreen Forest, and Shrubland/Herbaceous classes that can be explained as the timber harvest and reforestation stages of forests managed for timber, many of which are pine plantations. Monoculture pine plantations are common in the southeastern U.S. and the final harvest is typically a clearcut, hence most of these changes are temporary land-cover changes that do not necessarily reflect long-term land-use changes. However, looking at net changes in each LULC class we see a dramatic decrease in Deciduous/Mixed Forest followed by Hay/Pasture and Evergreen Forest, respectively, while there is a considerable increase in the Barren and Urban classes as a result of mining activities, road construction and transportation infrastructure establishment, and urban expansion (Tables 2 and 3). See the figures in the Supplementary for examples of these land-cover categories.

**Table 2.** Changes in LULC classes from 2011 to 2019.

| LULC Class | Loss | Gain | Net Change | Net Change |
|---|---|---|---|---|
| | Area (ha) | Area (ha) | Area (ha) | % |
| Water | −405 | 710 | 305 | 4 |
| Urban | −23 | 10,643 | 10,620 | 12 |
| Barren | −638 | 1154 | 516 | 18 |
| Deciduous/Mixed Forest | −26,176 | 15,884 | −10,292 | −6 |
| Evergreen Forest | −36,381 | 31,980 | −4401 | −3 |
| Shrubland/Herbaceous | −37,582 | 44,160 | 6578 | 9 |
| Hay/Pasture | −8863 | 4198 | −4665 | −6 |
| Cultivated Crop | −1215 | 1243 | 28 | 0 |
| Woody Wetlands | −488 | 1800 | 1311 | 2 |

**Table 3.** Annual average of land-use exchanges within the UFW in an 8-year time interval from 2011 to 2019; change from (rows) to (columns) where negative and positive values represent losses and gains, respectively. Amounts represented are average annual changes in the area of each land use, measured in hectares.

| | Water | Urban | Barren | Deciduous/Mixed Forest | Evergreen Forest | Shrubland/Herbaceous | Hay/Pasture | Cultivated Crop | Woody Wetlands |
|---|---|---|---|---|---|---|---|---|---|
| Water | 0 | 3 | −5 | −1 | 0 | −14 | −1 | 0 | −16 |
| Urban | −3 | 0 | −17 | −292 | −254 | −169 | −397 | −40 | −7 |
| Barren | 5 | 17 | 0 | −6 | −33 | −38 | −2 | −1 | 0 |
| Deciduous/Mixed Forest | 1 | 292 | 6 | 0 | 556 | 208 | −45 | 28 | 99 |
| Evergreen Forest | 0 | 254 | 33 | −556 | 0 | 829 | −86 | −13 | 27 |
| Shrubland/Herbaceous | 14 | 169 | 38 | −208 | −829 | 0 | 19 | 47 | 19 |
| Hay/Pasture | 1 | 397 | 2 | 45 | 86 | −19 | 0 | −10 | 16 |
| Cultivated Crop | 0 | 40 | 1 | −28 | 13 | −47 | 10 | 0 | 7 |
| Woody Wetlands | 16 | 7 | 0 | −99 | −27 | −19 | −16 | −7 | 0 |
| Total | 34 | 1180 | 57 | −114,356 | −489 | 731 | −581 | 3 | 146 |

Reviewing the exchanges between land-use classes, we see that changes from all other land uses to urban areas tend to be more permanent. In other words, there seem to be lower possibilities for conversion of urban areas to other land-use classes. Hay/Pasture, Deciduous/Mixed Forest, Evergreen Forest, and Shrubland/Herbaceous classes contributed to increases in urban area (Figure 2).

We identified dominant land-use transitions in the UFW, categorized under Urbanization, Forestry Practices, Afforestation, and Deforestation sub-models, in order to inform the MLP module. After Forestry Practices, Urbanization is the most common form of land-use transition, followed by conversion of forestlands to land uses other than urban areas. Figure 3 portrays major land-use changes with an area over 20 hectares in the UFW.

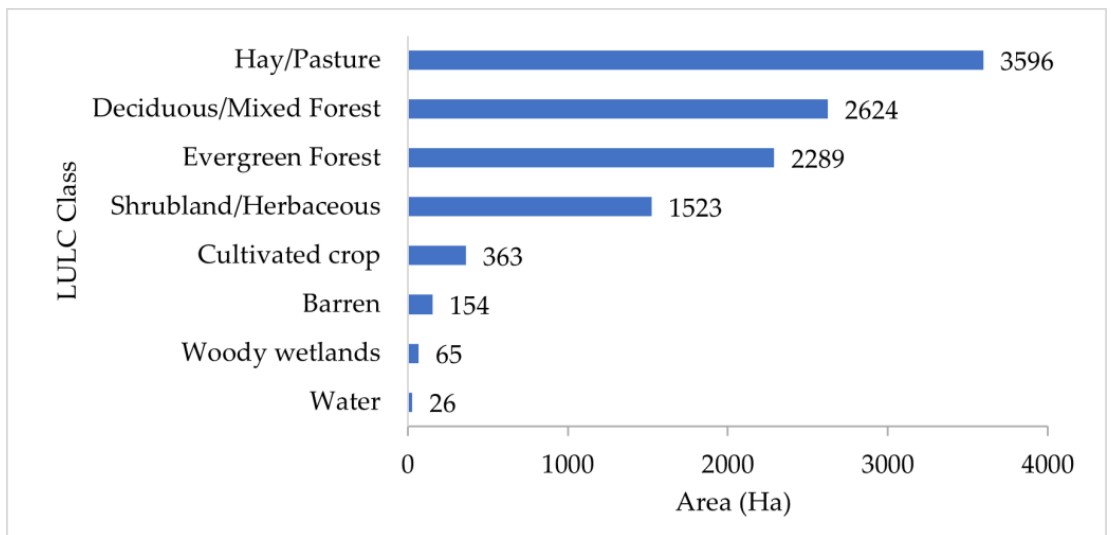

**Figure 2.** The total contribution of various land uses to urban area expansion, 2011–2019.

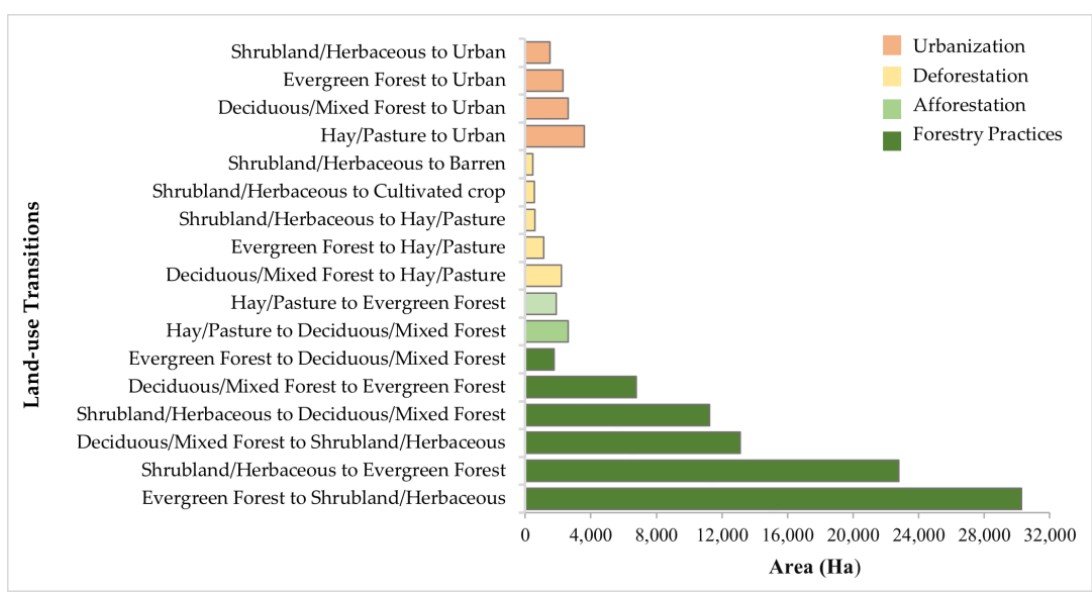

**Figure 3.** Major land-use transitions within the UFW, 2011–2019.

*5.2. Validation Results*

Overall, Kappa indices for the projected maps using the CI layer and with no CI layer for LULC 2019 are 0.97 and 0.95, respectively. Tables 4 and 5 represent Kappa indices for separate categories along with Quantity Disagreement, Allocation Disagreement, and Total Disagreement for projected maps. The error matrices of the projected LULC map without using the CI layer, and with the CI layer against the NLCD 2019 map as the true reference image, are provided in the Supplementary.

**Table 4.** Accuracy assessment results for the projected LULC map of 2019 without using the CI layer. QD = Quantity Disagreement; AD = Allocation Disagreement.

| LULC | Kappa | QD | AD | Total Disagreement |
|------|-------|-----|-----|--------------------|
| Water | 0.99 | 0.00 | 0.01 | 0.01 |
| Urban | 0.98 | 0.01 | 0.02 | 0.02 |

**Table 4.** *Cont.*

| LULC | Kappa | QD | AD | Total Disagreement |
|---|---|---|---|---|
| Barren | 0.63 | 0.25 | 0.12 | 0.37 |
| Deciduous/Mixed Forest | 0.95 | 0.00 | 0.05 | 0.05 |
| Evergreen Forest | 0.90 | 0.01 | 0.09 | 0.10 |
| Shrubland/Herbaceous | 0.66 | 0.00 | 0.33 | 0.33 |
| Hay/Pasture | 0.99 | 0.00 | 0.01 | 0.01 |
| Cultivated Crop | 0.99 | 0.00 | 0.01 | 0.01 |
| Woody Wetlands | 1.00 | 0.00 | 0.00 | 0.00 |
| Total | 0.95 | 0.27 | 0.64 | 0.91 |

**Table 5.** Accuracy assessment results for the projected LULC map of 2019 using the CI layer. QD = Quantity Disagreement; AD = Allocation Disagreement.

| LULC | Kappa | QD | AD | Total Disagreement |
|---|---|---|---|---|
| Water | 0.99 | 0.00 | 0.01 | 0.01 |
| Urban | 0.98 | 0.02 | 0.00 | 0.02 |
| Barren | 0.63 | 0.30 | 0.07 | 0.37 |
| Deciduous/Mixed Forest | 0.97 | 0.00 | 0.03 | 0.03 |
| Evergreen Forest | 0.95 | 0.03 | 0.05 | 0.08 |
| Shrubland/Herbaceous | 0.74 | 0.05 | 0.20 | 0.25 |
| Hay/Pasture | 1.00 | 0.00 | 0.00 | 0.01 |
| Cultivated Crop | 1.00 | 0.01 | 0.01 | 0.01 |
| Woody Wetlands | 1.00 | 0.00 | 0.00 | 0.00 |
| Total | 0.97 | 0.41 | 0.37 | 0.78 |

*5.3. Projected LULC 2040*

We projected year 2040 LULC maps under Business as Usual, Urbanization, Conservation, and Maximum Forest Protection scenarios (Table 6). To do this, we incorporated socio-economic and policy factors in the LCM module by multiplying the CI layers of each development scenario to calculate the possibility of land-use transitions under each sub-model combined with MC values (Figure 4).

**Table 6.** Total area projections in hectares for different LULCs in 2040 compared to 2001 and 2009 area calculations.

| | NLCD 2001 | NLCD 2019 | Projected LULC 2040 | | | |
|---|---|---|---|---|---|---|
| | | | Business as Usual | Urbanization | Conservation | Maximum Forest Protection |
| Water | 7552 | 8171 | 8310 | 8310 | 8310 | 8310 |
| Urban | 73,393 | 86,548 | 101,039 | 104,202 | 99,334 | 91,805 |
| Barren | 2537 | 2862 | 3257 | 3232 | 2875 | 2758 |
| Deciduous/Mixed Forest | 191,147 | 181,369 | 171,426 | 170,689 | 173,193 | 175,422 |
| Evergreen Forest | 183,350 | 170,777 | 172,109 | 171,067 | 172,621 | 176,177 |
| Shrubland/Herbaceous | 51,519 | 70,888 | 74,356 | 73,747 | 74,973 | 75,830 |
| Hay/Pasture | 94,288 | 83,108 | 72,528 | 72,278 | 73,176 | 73,176 |
| Cultivated Crop | 20,501 | 20,722 | 20,922 | 20,922 | 20,963 | 20,964 |
| Woody Wetlands | 56,780 | 56,621 | 56,618 | 56,618 | 56,623 | 56,624 |

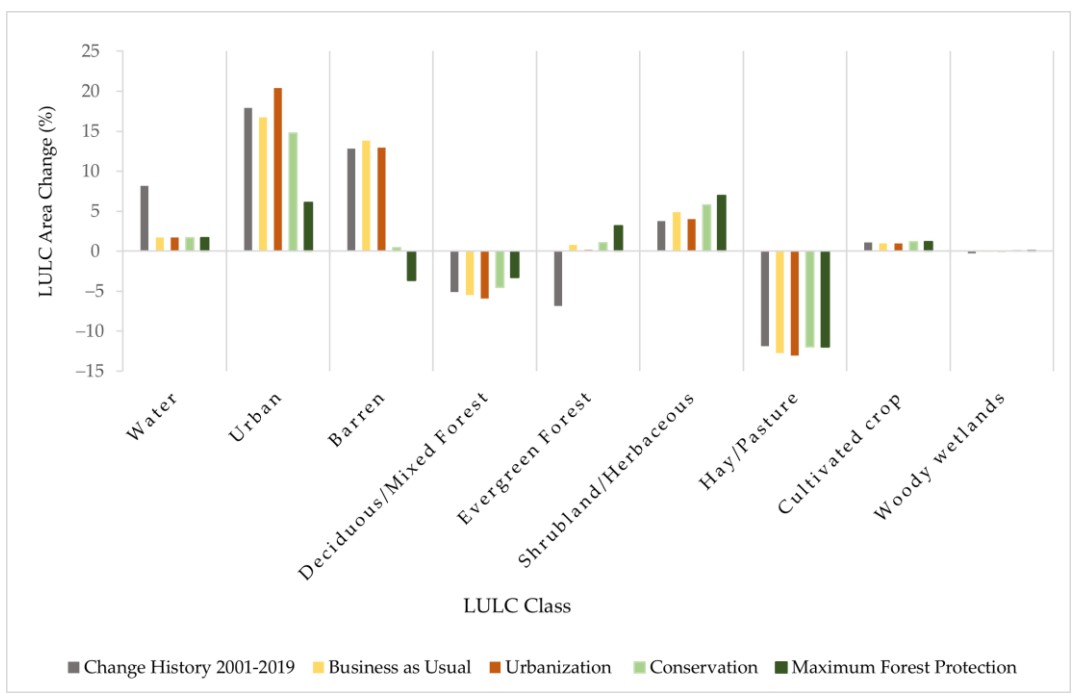

**Figure 4.** Percentage of change in total area of target LULCs under various 2040 development scenarios.

Focusing on Deciduous Forest, Evergreen Forest, Shrubland/Herbaceous, Urban, and Hay/Pasture as land uses with considerable changes, we expect a decline in the Pasture and Deciduous/Mixed Forest classes under all 2040 development scenarios. The total area of Evergreen Forest is expected to remain roughly the same or to experience an increase under all the future scenarios (the decline in the 2001–2019 period was largely due to the establishment of Lake McIntosh near Peachtree City).

## 6. Discussion

In this research, we chose to use NLCD maps for 2011 and 2019 to study dominant LULC changes, allowing the models to avoid being overly influenced by occasional massive anthropogenic changes such as the McIntosh Dam establishment or the Atlanta Hartsfield-Jackson expansion project that happened in earlier periods. We also used NLCD maps of 2011, 2016, and 2019 to compare observed changes in our projected LULC map for 2019 and validate our model.

Comparing the model validation results, we found that incorporating socio-economic and policy factors through the CI layer helped to improve the projection approach, decreased total disagreements among classes, and slightly improved the Kappa indices for LULCs. However, there remain considerable false projections among Shrub/Herbaceous and Barren land as these two LULC classes are the most transient land uses in the region due to prevailing forestry practices as well as activities such as solar panel installment, establishment of road construction projects or mining activities. Moreover, land-use classes with similar reflectance in satellite images may lead to classification confusion among some LULC classes, hence there are classification errors associated with these two land uses that can negatively affect the overall accuracy level of projected maps [56,76,77].

The LULC transition analysis shows that Barren, Urban, and Shrubland/Herbaceous categories are increasing over time while Deciduous/Mixed Forest, Pasture, and Evergreen Forest are declining. However, the apparent decline in Evergreen Forest and the increase in Shrubland/Herbaceous is likely due to the timber harvest and re-establishment processes typical of the southeastern U.S. Forest harvest and urbanization are the predominant LULC change drivers in the UFW.

Projected LULC maps of the UFW for the year 2040 under all of our hypothetical development scenarios anticipate consistent expansion of urbanized area and consequently a dramatic decline in the Deciduous/Mixed Forest category. Deciduous/Mixed Forest is also losing ground via conversion to pure Evergreen Forest. At the same time, we anticipate a constant increase in the Evergreen Forest and Shrubland/Herbaceous categories under all development scenarios except for the Urbanization scenario, suggesting that the total area of Evergreen Forest will remain relatively constant even under high urbanization pressures.

A comparison of our Business as Usual and Urbanization scenarios reveals that the decline in Deciduous/Mixed Forest is almost the same in both scenarios, while under the Business as Usual scenario we can expect an increase in Evergreen Forest. This indicates that current conservation policies, mostly voluntary incentive programs, are more effective in protecting the more commercially valuable Evergreen Forest than the potentially more ecologically valuable Deciduous/Mixed Forest [56]. Comparing the Maximum Forest Protection scenario with other development scenarios, we see that Deciduous/Mixed Forest continues to decline at a slower pace while Evergreen Forest increases dramatically. Small (less than 20 ha) parcels of Deciduous/Mixed Forest located in proximity to dense developed areas are the most likely to be converted to urban areas as a result of urban growth in rural areas, while the loss of larger Deciduous/Mixed Forest parcels in rural areas farther from metropolitan areas is due to motivations for maximizing revenue from forestry practices through conversion to intensive pine plantations.

Under the Urbanization scenario with a loosening of current conservation policies and higher market pressure promoting development, we see a lower growth rate in Evergreen Forest when compared to other scenarios. Evergreen Forest continues to expand, however it does so at a lower rate than the loss of Deciduous/Mixed Forest. Loss of the latter category is pronounced in the upper parts of the UFW. In other scenarios the total area of Evergreen Forest remained constant or increased. The lower rate of Evergreen Forest expansion in the upper parts of the UFW under the Urbanization scenario is likely due to parcelization and the fact that forestry practices in small parcels cannot compete with the high development market pressures within proximity to the AMA due to economies of scale. Our findings are aligned with other studies focused on Georgia's forest cover transitions suggesting that Evergreen Forest is increasing in Georgia's exurban and rural areas but decreasing in proximity to urban areas [7,10,47,78,79].

Although our four scenarios provide marked contrasts with one another, all four are informed by past patterns of LULC change. Novel policy or socio-economic dynamics could alter trajectories in the UFW in ways that are currently difficult to anticipate. For example, to date, Georgia forest owners have been relatively inactive in carbon offset markets, but changes in market conditions or in climate policy could affect landowner decision-making in ways that result in greater forest conservation or establishment. Likewise, Woody Wetlands showed relatively little change under any scenario here, but very recent changes in federal Clean Water Act interpretation could potentially result in the loosening of restrictions on conversion of these valuable ecosystems to other land uses. At the same time, sudden shifts or "tipping points" in climate feedbacks could make some current land uses economically and ecologically inviable and open the possibility to expand other, currently inviable, uses.

## 7. Conclusions

This work demonstrated that deforestation as a result of urban expansion can be quantitatively projected to a future time point using the LCM model with reasonable accuracy. Previous research on projection approaches has also suggested that the LCM is suitable for simulating new development in regions with urban densities of less than 80 percent [74]. While our accuracy assessments represented an acceptable level even without using the CI layer, the integration of socio-economic and policy factors through the CI layer along with biophysical variables helped to include random, chance events of new development far from existing urban areas and to better anticipate the trajectory

of land-use changes in urbanized watersheds. This complements previous studies that emphasize the importance of incorporating socio-economic variables [27,42,80,81] and the proximity of roads [78], rivers, and areas with flat topography [10,80] to project complex human-natural dynamics. However, based on the findings of this study, incorporating socio-economic and policy factors into projection approaches adds more complexity to projection models while adding only modest improvements in accuracy. Given the tradeoffs between model complexity and errors, and the fact that adding details to a model does not always guarantee an increase in its reliability [17], we believe users should rely on their understanding of their study regions' anthropogenic changes, available effective policies, and demographic factors when deciding upon the level of desirable complexity in their projection method.

Projecting LULC under various development scenarios helps decision-makers and stakeholders explore the outcomes of different levels of change in demographic factors or conservation policies. For instance, comparing results of the Business as Usual and Urbanization scenarios showed that under a constant population growth rate, a slight loosening of current conservation strategies accompanied by market pressures can lead to excessive loss of forestlands, especially the valuable Deciduous/Mixed Forest category and its numerous ESs. Despite the higher weight given to development constraint factors in the Conservation scenario, we still see a constant decline in Deciduous/Mixed Forest. This points to the need for more effective conservation strategies and policy tools if maintaining the same level of available ESs in the future is desired, especially in watersheds with higher urbanization pressures [10,79,82].

Comparing the LULC projection model validation results showed that including LULC change driving forces via the CI layer helps to produce more reliable LULC projections. In addition, this approach provides the opportunity to simulate and visualize the outcomes of various hypothetical development scenarios that can be useful in development planning, natural resources management, risk assessment and management, policy design, and decision-making processes. Our findings are aligned with previous studies emphasizing the importance of including socio-economic and policy factors in LULC projection approaches [10,42,79,82].

## 8. Limitations and Uncertainties

In regions with extensive areas under intensive forestry practices, such as pine plantations in the southeastern U.S., land-cover changes due to timber harvest and replantation processes can cause misclassifications and reduce the overall accuracy level of projection analysis. All of the data we used in this study are publicly available for the entire conterminous United States, which makes this projection approach replicable in other regions within the U.S. However, LULC change forces vary substantially across regions and users may need to modify the approach based on local natural and anthropogenic factors, economic activities, and state- to local-level policies.

**Supplementary Materials:** The following supplementary information can be downloaded at https://www.mdpi.com/article/10.3390/su151914270/s1. Table S1. LULC Class definitions and reclassification IDs (source: https://www.mrlc.gov/data, accessed on 27 February 2021); Table S2. Species listed as federally threatened or endangered under the Endangered Species Act within the UFW as of 2022; Table S3. An example of Markov chain matrix used for model validation using NLCD2011 and NLCD 2016; Table S4. An example Markov chain matrix used for projecting LULC 2040 generated based on NLCD 2011 and NLCD 2019; Table S5. Error matrix of the projected LULC without the CI layer (rows) against the NLCD 2019 map (columns) as the reference image; Table S6. Error matrix of the projected LULC using the CI layer (rows) against the NLCD 2019 map (columns) as the reference image; Figure S1. Urban sprawl in Deciduous/Mixed forests in the UFW, GA; Figure S2. Solar panel farms in the UFW, GA; Figure S3. Kaolin Mines within the UFW, GA; Figure S4. The Flint River, GA; Figure S5. Deforestation due to road development in the UFW, GA; Figure S6. Deforestation due to road development in the UFW, GA. References [83–97] are cited in supplementary materials.

**Author Contributions:** Conceptualization, B.A., J.B.A. and J.H.-C.; methodology, B.A., J.B.A. and J.H.-C.; software, B.A.; validation, B.A., J.B.A. and J.H.-C.; formal analysis, B.A.; investigation, B.A.; resources, B.A. and J.B.A.; data curation, B.A.; writing—original draft preparation, B.A.; writing—review and editing, B.A., J.B.A. and J.H.-C.; visualization, B.A.; supervision, J.B.A. and J.H.-C.; project administration, J.B.A.; funding acquisition, J.B.A. All authors have read and agreed to the published version of the manuscript.

**Funding:** This research was funded by USDA National Institute of Food and Agriculture McIntire Stennis project 1025989.

**Institutional Review Board Statement:** Not applicable. This study did not involve any human or animal.

**Informed Consent Statement:** Not applicable.

**Data Availability Statement:** Publicly available data can be acquired through: https://www.usgs.gov/centers/eros/science/national-land-cover-database, (accessed on 27 February 2021) https://www.census.gov/data.html (accessed on 24 September 2022).

**Conflicts of Interest:** The authors declare no conflict of interest. The funders had no role in the design of the study; in the collection, analyses, or interpretation of data; in the writing of the manuscript; or in the decision to publish the results.

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
