# Peer review of "Incorporating Social and Policy Drivers into Land-Use and Land-Cover Projection"

_sustainability, doi:10.3390/su151914270_

Round 1

Reviewer 1 Report

In introduction chapter it would be good to explain the role of LULCF mitigation potential regarding mitigation of climate change (IPCC report for the agriculture forestry and  other land use,  Sustainable development goals regarding ecosystem services).

In line 90, subchapter 2.1 authors explain possible loss od ecosystem services. Please explain which are the most important or threatened ES in the research area.

In discussion chapter please explain how LULC projection model validation results influence on the forest management in protected area. Who are the main stakeholders influenced with different scenarious, and how future climate change can influence on conversion to pure Evergreen Forests. Monoculture stands are under the different scenario analysis, what cause different Criteria and Indicators in the future for forest risk management and ecosystem services modelling.

Author Response

We sincerely thank the reviewers for their helpful comments and suggestions. Our detailed responses are provided below.

Reviewer 1:

  • In the introduction chapter, it would be good to explain the role of LULCF mitigation potential regarding climate change mitigation (IPCC report for agriculture forestry and other land use, Sustainable development goals regarding ecosystem services).

Response: We added lines 34 to 41 to detail the impact of LULC changes in terms of disturbing the biogeochemical balances between atmosphere and land at global scale:

Land cover and land management changes affect exchanged energy, water, aerosol, and greenhouse gas (GHG) fluxes between the land and atmosphere [2]. Alteration of land use and land cover (LULC) patterns can significantly affect natural ecosystem functioning and the ecosystem services (ESs) they provide to human societies, not only at a local scale, but also through GHG fluxes and changes in radiative transfer contributing to changes in atmospheric chemistry and thermal and moisture balance at global scales [2].

  • In line 90, subchapter 2.1 authors explain possible loss od ecosystem services. Please explain which are the most important or threatened ES in the research area.

Response: As we intended to keep the background session mostly related to previous research, literature reviewed and talking about broad general issues and ideas, we added a paragraph under the study area session to describe those ESs that concerned us the most in the Upper Flint Watershed. We added the following text:

The Flint River and its tributaries provide numerous provisioning ESs such as water for agricultural, industrial, and municipal uses along with varied regulating, supporting, and cultural ESs; an example of the latter is recreational opportunities for local people. The watershed is home to a diverse population of flora and fauna including several species that are listed as endangered or threatened.

  • In the discussion chapter please explain how LULC projection model validation results influence on the forest management in protected area. Who are the main stakeholders influenced with different scenarios, and how future climate change can influence on conversion to pure Evergreen Forests. Monoculture stands are under the different scenario analysis, what cause different Criteria and Indicators in the future for forest risk management and ecosystem services modelling.

Response: While we appreciate this comment, fully addressing it would add substantial length to the manuscript and is substantially outside the scope of the paper. However, we did include an entirely new paragraph at the end of the Discussion section that at least partially addresses these points:

Although our four scenarios provide marked contrasts with one another, all four are informed by past patterns of LULC change. Novel policy or socio-economic dynamics could alter trajectories in the UFW in ways that are currently difficult to anticipate. For example, to date Georgia forest owners have been relatively inactive in carbon offset markets, but changes in market conditions or in climate policy could affect landowner decision-making in ways that result in greater forest conservation or establishment. Likewise, woody wetlands showed relatively little change under any scenario here, but very recent changes in federal Clean Water Act interpretation could potentially result in the loosening of restrictions on conversion of these valuable ecosystems to other land uses. At the same time, sudden shifts or “tipping points” in climate feedbacks could make some current land uses economically and ecologically inviable and open the possibility to expand other, currently inviable, uses.

Reviewer 2 Report

Assigning the factors for different drivers of land use change in the study basin was interesting as a way of incorporating social and policies into Land Use projection. I found the authors explained completely and thoroughly about their methodology. 

Author Response

We sincerely thank the reviewers for their helpful comments and suggestions. Our detailed responses are provided below.

Reviewer2:

  • Assigning the factors for different drivers of land use change in the study basin was interesting as a way of incorporating social and policies into Land Use projection. I found the authors explained completely and thoroughly about their methodology. 

Response: We Highly appreciate your comment on our methodology.

Reviewer 3 Report

This manuscript incorporates social and policy drivers into land use and land cover projections. The background and process of the study are described in detail, and the work is somewhat novel. Below are my suggestions for improvement:

1. The analysis of ecosystem services in the title is not reflected in the text.

2. The abstract suggests additional data to support the conclusions.

3. References should be cited in the text in numbered order from smallest to largest.

4. Explain why it is a projection of LULC in 2040.

5. Check table 4, the line numbers are shown inside the table.

6. Figures 2, 3 and 4 are missing the x-axis and the font colour should be standard black.

7. The conclusion is suggested to be abbreviated, some parts can be moved to the discussion.

8. Many references have incomplete information or the year of publication is repeated, please check and correct them in accordance with the uniform format of the journal.

9. There are too many references, it is suggested to delete some non-essential citations.

Author Response

We sincerely thank the reviewers for their helpful comments and suggestions. Our detailed responses are provided below.

Reviewer 3:

  • The analysis of ecosystem services in the title is not reflected in the text.

Response: Ecosystem Services analysis typically use LULC maps as the base layer for computational calculations. Hence, to determine to what extent anthropogenic changes might affect the availability of ESs in the future, the first step is to project LULC for that time frame. We agree that including the term “Ecosystem Services Analysis” in the title might be confusing, hence we omitted the term from the title.

  • The abstract suggests additional data to support the conclusions.

Response: Unfortunately, this comment does not provide enough detail for us to offer a specific revision or explanation in response.

  • References should be cited in the text in numbered order from smallest to largest.

Response: We have updated our citation management software so that references are appropriately numbered.

  • Explain why it is a projection of LULC in 2040.

Response: We added the following text under section 4.1 to fulfill this point:

Georgia Regional Commissions have prepared their future land management and development plans; hence we utilized the same time frame to incorporate local governments’ strategic planning timeline into our model.

  • Check table 4, the line numbers are shown inside the table.

Response: Thank you; we have fixed this issue.

  • Figures 2, 3 and 4 are missing the x-axis and the font colour should be standard black.

Response: We have fixed these issues.

  • The conclusion is suggested to be abbreviated; some parts can be moved to the discussion.

Response: We removed a paragraph to avoid having a lengthy conclusion.

  • Many references have incomplete information or the year of publication is repeated, please check and correct them in accordance with the uniform format of the journal.

Response: We have reviewed the references and fixed these issues.

  • There are too many references, it is suggested to delete some non-essential citations.

Response: We agree that the manuscript contains a long list of references. Almost 20 of them are from the appendix (we were advised to keep references for the appendix and main body together) and 6 of them are related to geospatial data sources. We eliminated 6 unnecessary references; however, we ended up adding new information and some new references to address other reviewers’ suggestions during the revision.

Reviewer 4 Report

This paper presents a well-structured and written assessment of land use change and predicted change in an area adjacent to a major metropolitan area in the U.S. I found the study was clear and easy to follow. I am not an expert in GIS application but I found few issues with this paper that require additional attention.

My interpretation is that some of the greatest projected losses in land cover types are projected for the 'Water' and 'Woody Wetlands' land cover types (row to column in Table 3). You mentioned the EPA and wetland protection in the Introduction but I did not see detail to wetland protection of loss in the Discussion.  consider adding detail to the Discussion on this. Since you brought the EPA up in the Intro it could be helpful to discuss these potential losses and conversions in the context of the wetland protections recently limited by the EPA.

Couple minor issues:

There is much use of acronyms in this paper, something that affects my ability to follow without looking back for clarification as to what the abbreviations represent. Please review and reconsider if all abbreviations are needed.

You mention Kappa on line 358 providing a citation to this work. The reference cited seems esoteric and is difficult to access. Since this is the first use of the Kappa statistic in the text, and this statistic has bearing on the assessment used in this study, please define what Kappa is and provide an additional citation towards its description.

Author Response

We sincerely thank the reviewers for their helpful comments and suggestions. Our detailed responses are provided below.

Reviewer 4:

This paper presents a well-structured and written assessment of land use change and predicted change in an area adjacent to a major metropolitan area in the U.S. I found the study was clear and easy to follow. I am not an expert in GIS application but I found few issues with this paper that require additional attention.

  • My interpretation is that some of the greatest projected losses in land cover types are projected for the 'Water' and 'Woody Wetlands' land cover types (row to column in Table 3). You mentioned the EPA and wetland protection in the Introduction but I did not see detail to wetland protection of loss in the Discussion.  consider adding detail to the Discussion on this. Since you brought the EPA up in the Intro it could be helpful to discuss these potential losses and conversions in the context of the wetland protections recently limited by the EPA.

Response: Overall, the amount of change to Water and Woody Wetlands in all four scenarios was quite small. However, in a new paragraph at the end of the Discussion section we refer to the possibility that policy changes could open the door to greater levels of Woody Wetland conversion:

Likewise, Woody Wetlands showed relatively little change under any scenario here, but very recent changes in federal Clean Water Act interpretation could potentially result in the loosening of restrictions on conversion of these valuable ecosystems to other land uses.

  • There is much use of acronyms in this paper, something that affects my ability to follow without looking back for clarification as to what the abbreviations represent. Please review and reconsider if all abbreviations are needed.

Response: Unfortunately, this is a common issue with scientific papers using various geospatial data and models. We removed two acronyms as part of the revision (Quantity Disagreement and Allocation Disagreement). We will reach out to the editors to see if we can add a glossary of abbreviated terms to help address this issue.

  • You mention Kappa on line 358 providing a citation to this work. The reference cited seems esoteric and is difficult to access. Since this is the first use of the Kappa statistic in the text, and this statistic has bearing on the assessment used in this study, please define what Kappa is and provide an additional citation towards its description.

Response: We added a sentence to briefly describe Kappa, and used the less esoteric Pontius and Millones (2011) paper as a reference.

Round 2

Reviewer 3 Report

I still recommend that the authors add the X-axis to Figures 2 and 3 to better read the article for the reader. The authors did a good job of addressing my other questions, and I am confident that this manuscript will meet the journal's publication requirements.

Author Response

Comment: I still recommend that the authors add the X-axis to Figures 2 and 3 to better read the article for the reader. 

Response: Thank you for your quick review of our submitted manuscript. We were not 100% clear on what your comment was asking us to do, as Figures 2 and 3 did already have x-axes. We added lines and hash marks to the x-axes in both figures in hopes that this satisfies the suggestion.